# A Study of the Mechanisms and Characteristics of Fluorescence Enhancement for the Detection of Formononetin and Ononin

**DOI:** 10.3390/molecules28041543

**Published:** 2023-02-05

**Authors:** Jinjin Cao, Tingting Li, Ting Liu, Yanhui Zheng, Jiamiao Liu, Qifan Yang, Xuguang Li, Wenbo Lu, Yongju Wei, Wenhong Li

**Affiliations:** 1Department of Environmental and Chemical Engineering, Hebei College of Industry and Technology, Shijiazhuang 050091, China; 2College of Chemistry and Material Science, Hebei Normal University, Shijiazhuang 050024, China; 3Key Laboratory of Magnetic Molecules and Magnetic Information Materials (Ministry of Education), School of Chemistry and Material Science, Shanxi Normal University, Taiyuan 030031, China

**Keywords:** isoflavone, formononetin, ononin, fluorescence property, cleavage reaction

## Abstract

In this work, the origins for the spectral difference between two isoflavones, formononetin (F) and ononin (FG), are revealed via a comparison study of the fluorescence molecular structure. The fluorescence enhancement of FG in hot alkaline conditions is reported for the first time. For F, there is almost no fluorescence under acidic conditions, but when the pH is >4.8, its fluorescence begins to increase due to the deprotonation of 7-OH. Under a pH between 9.3 and 12.0, the anionic form of F produces a strong and stable fluorescence. The fluorescence quantum yield (Yf) of F is measured to be 0.042. FG shows only weak fluorescence in aqueous solutions under a wide range of pH until it is placed in hot alkaline solutions, which is attributed to the cleavage reaction of the γ-pyrone ring in FG. The Yf of FG is determined to be 0.020. Based on the fluorescence sensitization methods of F and FG, the quantitative analysis and detection of two substances can be realized. The limit of the detections for F and FG are 2.60 ng·mL^−1^ and 9.30 ng·mL^−1^, respectively. The linear detection ranges of F and FG are 11.7~1860 ng·mL^−1^ and 14.6~2920 ng·mL^−1^, respectively. Although the structural relationship between F and FG is glycoside and aglycone, under hot alkaline conditions, the final products after the cleavage and hydrolysis reactions are essentially different. The different fluorescence characteristics between F and FG pave a way for further identification and a quantitative analysis of the corresponding components in Chinese herbal medicine.

## 1. Introduction

Isoflavonoids are a class of compounds derived from isoflavone (3-phenylchromone), which widely exist in foods and drugs [1,2]. They are also the active ingredients in many traditional Chinese medicines (TCM). Formononetin (F) and ononin (FG) are a pair (glycoside and aglycone) of isoflavones (Figure 1) and are the two major flavones found in various Chinese herbal medicines, such as astragali radix (the root of *Astragalus membranaceus var. mongholicus* or *A. mem-branaceus*) [3,4] and red clover (*Trifolium pratense* L.) [5,6]. F and FG have many pharmacological effects; for example, antineoplasm [6,7,8,9], antioxidation [10], anti-inflammation [11,12], neuroprotection [12,13], and blood-lipid regulation [14,15,16]. Generally, the content of F and FG in TCM or biological samples is measured by liquid chromatography combined with absorbance and fluorescence or a mass spectrometry detector [17,18,19,20].

Fluorescence analysis has been widely used in biochemical analysis and chemical drug analysis [21,22], which is mainly because of its advantages, such as high sensitivity, high selectivity, simplicity, rapidity, and environmental friendliness. However, due to the complexity of the components contained in a sample and the lack of standard fluorescence spectral reference of the components in traditional Chinese medicine, the application of fluorescence analysis is not widespread for the analysis of traditional Chinese medicine. The difficulty is in determining how to avoid the interference from coexisting components when we determine the content of traditional Chinese medicine components using a fluorescence-based method. As such, it is of interest to study whether fluorescence spectroscopy, known for its inherent selectivity, can be used as an orthogonal detection technique for determining the ingredients in traditional Chinese medicine.

Previously, we studied the fluorescence properties of some flavonoids and found that some flavones and isoflavones underwent fluorescence enhancement reactions under hot alkaline conditions, which is attributed to the cleavage reaction in the C ring [22,23,24]. Eva de Rijke et al. observed that formononetin and ononin showed similar fluorescent emission [21]. They studied the effects of pH and solvents on the fluorescence of F and FG, and LC with fluorescence and MS detectors were used to separate and identify the impurities based on the shifts between *λ*_ex_ and *λ*_em_ of the fluorescent isoflavones. However, the report did not discuss the relationship between the photophysical properties and molecular structures of FG and F in-depth. The chemical changes of biomolecules and the relationship between their molecular structures and spectral characteristics have always been an important and interesting topic [25,26]. These spectroscopic studies and results have expanded the application and analysis scope of biomolecules to a certain extent.

In this paper, based on the previous research results, the fluorescence properties of F and FG and the relationship with the molecular structure were studied, and the conditions for selective fluorescence emission of the two were obtained. Focusing on the effect of pH on fluorescence behavior, the authors studied the mechanism of the C ring cleavage reaction in hot alkaline conditions and the fluorescence spectrum of their reaction products. The possible forms (molecular form, ionic form, and cleavage product) of the two compounds were explained based on the intrinsic characteristics of the molecular structure and the spectral information, including fluorescence wavelength and fluorescence intensity under different experimental conditions. The results showed that the molecular form of F did not fluoresce in the aqueous solution and the ionic form exhibited strong fluorescence due to the deprotonation of 7-OH under weak alkaline conditions. FG in aqueous solution was weakly fluorescent, but could produce strong fluorescence and a characteristic emission spectrum by the C ring cleavage reaction in a strong alkaline solution. These results provide new fundamental understandings for establishing fluorescence analysis methods of F and FG and pave a new way to expand the fluorescence analysis of isoflavones.

## 2. Results

### 2.1. Adsorption and Fluorescence Spectra of F Aqueous Solution

UV absorption spectra of F under different pHs were measured and the results are shown in Figure 1. Within the pH range of 2.2~11.2, as shown in Figure 1a, three absorption peaks appeared in the absorption spectra, which were located at 256, 303, and 334 nm. In this way, three isochromatic points located at 245, 285, and 312 nm were formed. The results suggest that the molecular form of F transformed into the ionic form under weak alkaline conditions. According to the pH-A data in Figure 1b, the acid dissociation constant of 7-OH (pK_a_) in F was 7.34 ± 0.01, which was calculated by pH-spectrophotometry. [27].

When pH ≥ 13.0, the absorption peaks at 334 nm showed no change, but the absorption peak at 256 nm decreased rapidly until it disappeared as shown in Figure 1c, which demonstrated that the structure in strong alkaline solutions was different from the ionic form. Figure 1d shows the comparison of the UV absorption spectra under three conditions, indicating that F in solution has three different structures under neutral, alkaline, and strong alkaline conditions.

The fluorescence spectra of F aqueous solution at different pHs were measured, as shown in Figure 2. F emitted weak fluorescence in the range of pH 2.0~5.0. The fluorescence peak of F was significantly enhanced at the excitation wavelength (λ_ex_) of 256 nm and 334 nm, and the emission wavelength (λ_em_) of 464 nm in weak acidic and weak basic conditions (pH > 4.8). Fluorescence reached maximum intensity, which preserved in pH range of 9.3~12.0, and then was quenched rapidly as the pH continued to increase (pH > 12.0) although the pattern of excitation and the emission spectra remained the same throughout the process. It was consistent with the trend of pH-dependent fluorescence intensity of F in the literature [21].

The reason why the F’s fluorescence spectra changed with pH is that the F molecule contains a hydroxyl group with an acidic proton. The hydroxyl group at position 7 (7-OH) in the F molecule is shown in Figure 1. The proton dissociation of 7-OH caused the change of the fluorescence spectrum when the pH of the solution changed from near neutral to weakly alkaline. The acidic dissociation constant of the 7-OH proton was determined to be pK_a_ = 7.31 ± 0.03 using the pH-F data shown in Figure 2d, which is based on the pH-Fluorescence method [28]. This pK_a_ value matches with the previous results obtained using pH-A.

In contrast to the isochromatic points in Figure 1a, there is no isofluorescence point in Figure 2, which suggests that both the molecular and ionic form of F can absorb light in the ultraviolet region, but only the ionic form can emit fluorescence.

As shown in Figure 3, it can be seen from the three-dimensional fluorescence spectra of F that the fluorescence was weak under neutral (Figure 3a) and alkaline conditions (Figure 3c), although F fluoresces strongly under weak alkaline conditions (Figure 3b, λ_ex_: 256 nm, 334 nm). This observation is consistent with the change in the UV absorption spectra (Figure 1d).

According to the consistent change rule of fluorescence spectra and UV absorption spectra of F, we speculate that when pH < 6.0, F mainly exists as a molecular form (Figure 2A); while under alkaline conditions (pH = 9.3~12.0), F loses a 7-OH proton to be converted to the ionic form (Figure 2B). F undergoes a pyran ring cleavage under strong alkaline conditions (pH > 13). The alkaline solution (pH = 13.5) containing compound C was placed at room temperature for 2 h and hydrochloric acid was added to adjust the pH of the solution (about pH = 13). The fluorescence spectra and intensity of the above solutions had no changes, indicating that once compound C is formed under certain conditions it cannot be converted to compound B even if it is adjusted to acidic conditions. This phenomenon is consistent with the mechanism reported in the literature [29]. Compound C was obtained and the molecular structure of C was further confirmed, as shown in Appendix A.

### 2.2. Fluorescence and Adsorption Spectrum of FG and Its Cleavage Product

Different from F, FG showed weak fluorescence under nearly neutral conditions (Figure 4b) and almost no fluorescence under acidic (Figure 4a) as well as alkaline conditions (Figure 4c). When the FG solution was under strong alkaline conditions (pH = 13.3) for 8 h at room temperature, there was still no obvious fluorescence emission in the scanning spectrum. The absence of ionizable hydroxyl protons and the presence of a 7-position oxoside (7-OGlu) prevented FG from producing fluorescence like F and thus, the change in pH could not affect the fluorescence spectrum of FG.

However, in the experiment we found that the FG aqueous solution would emit strong fluorescence when heated under alkaline conditions and the maximum excitation peak and emission peak (λ_ex_/λ_em_) was 288 nm and 388 nm, respectively, as shown in Figure 4d.

The fluorescence changes of the FG-derived fluorescent types under different pH conditions were systematically studied. The fluorescence intensity of the FG aqueous solution was still very weak after the solution was heated under a pH range of 1.0~8.8. However, when pH > 8.8, the fluorescence peak located at λ_ex_/λ_em_ = 243 nm/388 nm enhanced significantly and the fluorescent intensity reached a maximum at pH 12.5, followed by dropping sharply as the pH increased (pH > 12.5).

As the experimental results are shown in Figure 4, we can conclude that a structural transformation of FG took place and new fluorescent species were produced under hot alkaline conditions. According to the molecular structure of FG, there are two possible reactions: one is the ring-opening reaction of the γ-pyrone ring as shown in Figure 3 and and the other one is the hydrolysis reaction of 7-OGlu.

We inferred which kind of chemical reaction happened in FG combining with the fluorescence characteristics of the reaction product. If the 7-oxo-glycosidic bond was hydrolyzed, the reaction product should be like the ionic form of F after deprotonation of 7-OH. As shown in Figure 2, the fluorescence *λ*_ex_ is located at 256 nm and 334 nm, and *λ*_em_ is located at 464 nm. However, Figure 5 shows that the *λ*_ex_ (243 nm, 288 nm) and *λ*_em_ (388 nm) of the actual product were significantly different from the peak position in Figure 2. These results indicate that no hydrolysis reaction occurred under hot alkaline conditions. Another hypothesis is that FG underwent a ring-opening reaction of the γ-pyrone ring. If this was the case, then the cleavage product should have a lower degree of conjugation in the molecular structure compared to the ionic form of F. Moreover, according to the theory of fluorescence, it should have shorter wavelength fluorescence peaks. This theoretical speculation is in agreement with the results shown in Figure 5. Therefore, we inferred that the ring-opening reaction of the γ-pyrone ring occurred under hot alkaline conditions and the cleavage product showed fluorescence.

In order to investigate the structural changes of FG at different pHs under heating conditions, the UV spectrum of FG was measured (Figure 6).

The UV absorption spectra of FG were collected under the same heating condition and approximate pH range as the fluorescence spectra above. In Figure 6a, when heated in a pH range of 2.4~8.2, the FG solution showed two absorption peaks at 250 and 298 nm. Therefore, the structure of FG is the molecular form in Figure 3A’. As the pH increased, the absorption peaks at 250 and 298 nm decreased simultaneously and the peaks were red-shifted. When pH > 11.6, the absorption peaks were at 272 and 350 nm. The appearance of an isochromatic point at 318 nm confirmed the occurrence of structural changes between the two forms of FG under alkaline heating conditions. Comparing Figure 5c and Figure 6b, above pH 12.5, the fluorescence of FG was quenched, but its absorption peaks did not change significantly. The results indicate that the glycoside bond was not dissociated from FG in a highly alkaline solution. Under heating in a strong alkaline solution, two samples of FG with a pH of 13.5 were obtained and one of the samples had hydrochloric acid added, adjusting the pH to 13.3. By comparing the two samples, the fluorescence intensity of FG with pH 13.3 was significantly increased. The above fluorescence behavior of FG, to some extent, clarified that the fluorescence-quenching of FG under strong alkaline conditions (pH > 12.5) is not due to the dissociation of the glycoside bond from its fluorescent form, but the comprehensive influence of the solvent environment.

As shown in Figure 6b, under strong alkaline conditions (pH = 13.7), the pyrolysis product of FG has two absorption peaks (272 nm, 350 nm) and F has only one absorption peak (334 nm). The significant difference between the absorption spectra of the two indicated that the glycoside bond of FG had not dissociated under the strong alkaline heating conditions. The results suggest the existence of the 7-position glycoside bond inhibits the progress of a ring-opening reaction to a certain extent. In 2018, under strong alkaline conditions, the cleavage product of ipriflavone was obtained and a structure similar to FG was confirmed by molecular characterization [30]. The mechanistic analysis of the degradation of isoflavone can also be found in the literature [24]. Compound **B’** was obtained and the molecular structure of **B’** was further confirmed, as shown in Appendix A.

In summary, the fluorescence properties of F and FG were essentially different although the two molecular structures are a conjugated pair of glycoside and aglycone. We prove that FG cannot be converted to F by cleavage of the glycosidic bond under hot alkaline conditions. Instead, the γ-pyrone ring cleavage reaction occurs first to produce fluorescence and then fluorescence-quenching occurs under strong alkaline conditions.

### 2.3. Effect of Heating Temperature on Fluorescence of FG

As shown in Figure 7, the fluorescence of FG was extremely weak in the strong alkaline solution (pH 12.5) at room temperature and the fluorescence of FG was enhanced by increasing the heating temperature, which indicated that the higher temperature could accelerate the cleavage reaction. When the temperature was higher than 70 °C, the fluorescence reached a maximum, suggesting completion of the cleavage reaction. The resulting fluorescence spectra are similar to those in Figure 5b,c and show no obvious changes in the temperature range of 70~100 °C. In the subsequent experiments, we chose a boiling water bath for heating in order to facilitate operation.

### 2.4. Effect of Solvent on Fluorescence of F

The effect of solvent (the volume fraction of methanol in aqueous solution) on the fluorescence of F was investigated under weak alkaline conditions (pH 9.3). The results in Figure 8 show that although the volume fraction of methanol in the aqueous solution was changed, the position of the fluorescence peaks was almost unchanged (Figure 8a) although had a significantly enhanced peak intensity (Figure 8b). Therefore, in this study, the volume fraction of methanol in the aqueous solution was controlled at 10% when we investigated the effect of other experimental conditions on the fluorescence. In addition, if the content of F in Chinese medicinal materials is relatively low, in order to improve the sensitivity of the fluorescence method, the content of methanol can be increased to 50%, namely, 1:1 methanol to water ratio (*v*/*v*). This is largely consistent with the results presented in the literature [21].

Different from F, with an increasing volume fraction of methanol in the solvent, the intensity of the emission spectrum of the FG solution increased (Figure 9a) while the maximum emission wavelength showed a significant blue shift (Figure 9b). The fluorescence intensity of the FG solution reached a maximum, which improved about five times that of methanol/water (*v*/*v*) with 1:9 when methanol/water (*v*/*v*) reached 8:2; while in the pure methanol solution, the fluorescence intensity decreased slightly. When methanol/water (*v*/*v*) was changed from 1:9 to 10:0 as the solvent, the maximum emission wavelength of FG shifted from 500 nm to 460 nm. We found that under the same concentration, the fluorescence intensity of the pyrolysis product of FG obtained under hot alkaline conditions was more than twice that of neutral FG when the solvent condition of methanol/water is 8:2. Moreover, compared with the structural analogue F, the fluorescence spectrum of FG pyrolysis products showed good characteristics.

### 2.5. The Stability of Fluorescence

The experiments revealed that the fluorescence of F was basically stable when placed in a weak alkaline solution at room temperature. There was no change when the solution was continuously irradiated by a xenon lamp.

The fluorescence intensity of FG increased with the extension of heating time. It became stable after 1.5 h and did not change significantly when continuously exposed to the xenon lamp for 200 s, indicating that the fluorescence properties were essentially stable.

### 2.6. Fluorescence Quantum Yield

The fluorescence quantum yield (Yf) was measured according to the reference method and we selected quinine sulfate (Yf = 0.55, λ= 313 nm in 0.05 M H_2_SO_4_) and L-tryptophan (Yf = 0.14, λ= 280 nm in water) as the standards [22]. During the test, the UV absorbance of all solutions was lower than 0.05 to avoid internal filtering. The Yf of F in a weak alkaline solution (pH 9.3) was measured to be 0.042 and that of the pyrolysis product of FG in an alkaline solution (pH 12.5) was 0.020. Although the Yfs are simply moderate, we can still analyze them using modern fluorescent instruments with high sensitivity.

### 2.7. Relationship between Concentration and Fluorescence Intensity

Based on the above experimental results, a series of solutions (pH = 9.3) containing different concentrations of F, from 11.7 ng·mL^−1^ to 1860 ng·mL^−1^, were prepared and their fluorescence spectra were collected, as shown in the inset in Figure 10 and Figure 11. A standard curve of fluorescence intensity, based on IF (λ_ex_/λ_em_ = 334 nm / 464 nm) versus concentration of F, was drawn as shown in Figure 10a. In the range of 11.7–1860 ng·mL^−1^, IF showed a linear relationship with the concentration of F, as shown in Figure 10b. The regression equation is IF = 10.62 + 2.19 c, with the correlation coefficient R^2^ = 0.9997 (n = 14). The blank control was scanned and the lowest limit of detection for F was found to be 2.60 ng·mL^−1^ (9.69 × 10^−9^ mol·L^−1^).

Similarly, Figure 11a,b show the standard curve of IF (λ_ex_/λ_em_ = 288 nm/388 nm) versus c_FG_. The standard samples with a fixed pH at 12.5 were prepared by heating the samples in a boiling water bath for 1 h and cooling to room temperature before the fluorescence spectra (as shown in Figure 11b) were measured. The results show that the linear relationship between IF and c_FG_ is good in the range of 14.6–2920 ng·mL^−1^, fitted by the regression equation IF = 29.88 + 0.60 c with R^2^ = 0.9971 (n = 15), and the lowest limit of detection for FG was found to be 9.30 ng·mL^−1^ (2.16 × 10^−8^ mol·L^−1^).

The fluorescence-based detection of F and FG was compared with different methods in the literature and the details are summarized in Table 1. Using fluorescence analysis, the limits of quantification (LOQ) and detection (LOD) were 117 ng mL^−1^ and 2.6 ng mL^−1^ for F and 14.6 ng mL^−1^ and 9.3 ng mL^−1^ for FG, respectively. Compared with previously reported chromatography-based methods, such as UPLC-DAD-MS, RPLC–UV, and UHPLC-UV-MS, fluorescence analysis, which can avoid the waiting time for separation, was simpler to operate and consumed less methanol solvent. Moreover, the fluorescence analysis outperformed other methods in terms of the linear range, detection limit, simplicity of the sample pretreatment, and the lower cost than that of the combined methods. The characteristic fluorescence of F and FG was determined by this method. The content of F and FG in some traditional Chinese medicines can be determined under weak alkaline conditions and strong alkali heating conditions, respectively. In addition, the detection limit of this method can be pushed to the nanogram level, which is more sensitive than other methods and is more suitable for the quantitative analysis of traditional Chinese medicine with a low content of active ingredients. Therefore, this method provides a new way for the qualitative and quantitative analysis of traditional Chinese medicine.

It can be seen from the above results that when a sample contains both F and FG, although the structures of the two compounds are similar, the difference in fluorescence between them can be used for quantitative analysis. One can keep the sample solution under weak alkaline conditions for the analysis and determination of F and then the sample solution can be placed under strong alkali heating conditions to determine the content of FG. Further experiments are needed to confirm feasibility, which is our next research focus.

## 3. Materials and Methods

### 3.1. Materials

The chemical reference substances of F (CAS no. 485-72-3, molecular formula: C_16_H_12_O_4_, molecular weight: 268.26) were purchased from the Chinese National Institute for the Control of Pharmaceutical and Biological Products (Beijing, China) and of FG (CAS no. 486-62-4, molecular formula: C_23_H_23_NO_5_, molecular weight: 430.4) were purchased from TianJin Yifang Technology Co., Ltd. (Tianjin, China). Both F and FG were dissolved in methanol (chromatographic grade, Tedia, Fairfield, OH, USA) to prepare stock solutions. Quinine sulphate (molecular formula: C_20_H_26_N_2_O_6_S, molecular weight: 422.5), which was produced at the Chemical Limited Company of the Bodi Plant (Tianjin, China), was diluted to 1.00 × 10^−^^5^ M with 0.1 M H_2_SO_4_ when it was used. L-tryptophan (molecular formula: C_11_H_12_N_2_O_2_, biochemical reagent, chromatographic grade, molecular weight: 204.33) was purchased from the Institute of Microbiology, Chinese Academy of Sciences (Beijing, China) and was dissolved in water to a concentration of 1.00 × 10^−^^4^ M. Britton–Robinson buffer solutions were produced by mixing phosphoric acid, boric acid, and acetic acid (each 0.02 M), and adjusted to the appropriate pH by the addition of 0.1 M NaOH solution. All the buffer chemicals were of analytical grade. The water used throughout the study was double-deionized and verified to be free from fluorescence.

### 3.2. Apparatus

Fluorescence measurements were performed on a Hitachi (Tokyo, Japan) F-7000 fluorescence spectrophotometer equipped with a xenon lamp and 1 cm quartz cell. The excitation and emission slits (band pass) 5 nm/5 nm were used. Absorption spectra were recorded using a Shimadzu (Kyoto, Japan) UV-2501PC recording spectrophotometer with 1 cm quartz cell. An Orion (Beverly, MA, USA) 868 pH/ISE meter was used for pH measurement. A five-digit analytical balance of up to 0.01 mg was used.

### 3.3. General Procedure for Spectral Measurement

Appropriate amounts of F or FG and buffer solutions were added into a series of 10 mL volumetric flasks. The mixtures were then diluted to the mark with methanol or water and mixed well. Lastly, the fluorescence or absorption spectra and pH were measured at room temperature.

### 3.4. Measurement of Fluorescence Quantum Yield

Quantum yield was estimated by a referential method [26], and quinine sulfate (quantum yield 0.58) and L-tryptophan (quantum yield 0.14) were used as references. For the measurement, quinine sulfate, L-tryptophan, and CH or ECH solutions were prepared at an appropriate concentration so that their absorbance (𝐴) was not larger than 0.05. The absorption and fluorescence spectra were recorded and the quantum yield was calculated according to Equation (1):(1)$$Y_{u}=\left[\left(F_{u}A_{s}\right)/\left(F_{s}A_{u}\right)\right]\times Y_{s}$$
where *Y*_u_ and *Y*_s_ were the fluorescence quantum yields of unknown; the references, *F*_u_ and *F*_s_, were the integral fluorescence intensities of unknown; and the reference solutions, *A*_u_ and *A*_s_, were the absorbance of unknown and reference solutions at their excitation wavelengths, respectively.

## 4. Conclusions and Outlook

In summary, F is weakly fluorescent under acidic and neutral conditions, and in weak base solutions, it can produce fluorescence at λ_ex_/λ_em_ = 334 nm/464 nm, which is due to the deprotonation of 7-OH. Under strong alkaline conditions, the cleavage of the γ-pyrrole ring leads to fluorescence-quenching. Since there are no dissociable protons in FG, its fluorescence in aqueous solutions is very weak under normal temperature conditions and regardless of the pH of the solution. However, under high temperature alkaline conditions, FG undergoes a γ-pyrone ring cleavage reaction, resulting in increased fluorescence with λ_ex_/λ_em_ = 288 nm/388 nm. Although the structural relationship between F and FG is glycoside and aglycon, based on our experiment results the glycoside cannot be converted into the aglycone and therefore, the fluorescence enhancement mechanisms of them are essentially different. The fluorescence difference between F and FG under different experimental conditions lays the foundation for future fluorescence-based quantitative analysis. From an analytical chemistry point of view, these large differences of λ_ex_/λ_em_ between the two compounds under different experimental conditions enable high selectivity, which is the basis for fluorescence quantitative analysis. In the future, we will use the characteristic fluorescence of F and FG to determine their level of content in traditional Chinese medicine, which will provide support for the quality evaluation of traditional Chinese medicine.

## Data Availability

All data generated or analyzed during this study are included in this published article.

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
