# Peer review of "A Study of the Mechanisms and Characteristics of Fluorescence Enhancement for the Detection of Formononetin and Ononin"

_molecules, 2023, doi:10.3390/molecules28041543_

Round 1

Reviewer 1 Report

The manuscript studied in details of the absorption and fluorescence spectra of formononetin and ononin under different pH, and also presented the possible reaction mechanisms to explain the different fluorescence behavior of the two compounds under various pHs. While the study may help to differentiate the two isoflavones, the novelty and significance of the work is not sigificant enough. Moreover,  the proposed reaction mechanism in Scheme  3 was not fully verfied by 1HNMR and HRMS data.  There was no HRMS evidence for the formation of B'.  The 1HNMR spectrum shown in Figure S1 is not "pure" enough to confirm the formaiton of the C'.  Therefore, I don't think the manuscript  is publishable in this jounal. 

Reviewer 2 Report

The paper contains new and in-depth results about fluorescence behavior for two flavonoids F and FG. However, real applications for the method and validation data were missed. Major revision is required.

1-     In abstract “The linear ranges of F and FG are 0.0117-1.86 μg·mL−1 and 0.0146-2.92μg·mL−1, respectively.” It is recommended to write the linear ranges in ng.mL-1  ; nanogram and not microgram.

2-     In introduction “Anti-neoplastic [6-9], antioxidative [10], anti-inflammatory [11,12],

neuroprotective [12,13], blood-lipid regulation [14-16] and other pharmacological effects

of F and FG have been reported. “

Modify the sentence structure. Very long. It should be stated as “ F and FG have many pharmacological effects i.e………………..

3-     The following article should be cited and discussed in the introduction in more details because the two investigated compounds were also reported and the effect of Ph on their native fluorescence were demonstrated

https://www.sciencedirect.com/science/article/pii/S000326700200630X  

Merits of the new study over the aforementioned article should be stated in the introduction or the discussion

4-     In 3.3 Effect of Solvent on Fluorescence of F

Although 50 % methanol provide higher sensitivity the authors choose 10 % only ????

It is well known that higher sensitivity is recommended for completing the study

Additionally, in the following paper. The same results were reported

Fluorescence excitation and emission spectra of (A) F and (B) FG in methanol:water (1:1 (v/v)) solutions at different pH values.

https://www.sciencedirect.com/science/article/pii/S000326700200630X#FIG2

in-depth discussion and illustrations are required her.

5-     Where is the application of the method for analyzing real sample ?????

Therefore, this method was to be useful for the quantitative and qualitative analysis of F and FG in traditional Chinese medicine under different conditions.

Evidences is required for the aforementioned statement

6-     Additionally, analytical method validation is missed also ?

7-     Difference in methodology between new study and old reported ones should also be highlighted and illustrated in table form. Additionally,  merits, demerits, and real application should be compared

8-     Did the authors tried other solvents than water/methanol ?

9-     It is recommended to make the units in table 1 in ng.mL-1  

10-  future plan should be provided

Best wishes

Round 2

Reviewer 1 Report

The manuscript can be publishable unless the following issue can be addressed. The main concern is the whether the proposed product C' is truely responsible for the observed fluorescene emission increase of FG under hot alkaline solutions. The 1HNMR spectrum of C' shown in Figure S1 with unidentified peaks  at 5.0 ppm (d, about 1H), 4.9 ppm (d, about 1H), 3.7 ppm (dd, about 1H), 3.4-3.5 (m, about 3H) and the intergrals of these peaks are comparable to the peaks of the assumed product C', which is unacceptable. Since the fluorescence spectrum is very sensitive to impurities, the authors should  at least purify C' by HPLC and provide a clean 1HNMR spectrum of C' and also fluorescence emission specturm of a pure C' sample to confirm that the increase of fluorescence emission shown in Figure 5b was indeed due to the formation of C'. Otherwise, the proposed reaction mechanism shown in Scheme 3 may not be right, which greatly affect the quality of this work.  

Reviewer 2 Report

the authors replied for all comments in a very professional way. they did their best for improving the quality of the paper.

in my opnion, the paper can be published in the current form 

best wishes  

Author Response

Thank you for accepting our revised manuscript.

Reviewer 3 Report

I read the revised manuscript and answers to the quarry of my comments.

I found the present manuscript appears good and now I recommend it for publication.

Author Response

(The authors gave the same response as above.)
